# Drug Delivery Technology to the CNS in the Treatment of Brain Tumors: The Sherbrooke Experience

**DOI:** 10.3390/pharmaceutics11050248

**Published:** 2019-05-27

**Authors:** David Fortin

**Affiliations:** Division of Neurosurgery and Neuro-Oncology, Department of surgery, Faculty of Medicine and Health Science, University of Sherbrooke, Sherbrooke, QC J1H-5N4, Canada; David.fortin@usherbrooke.ca; Tel.: +1-819-346-1110 (ext. 73324)

**Keywords:** blood-brain barrier, intra-arterial chemotherapy, malignant gliomas, primary central nervous system lymphomas

## Abstract

Drug delivery to the central nervous system (CNS) remains a challenge in neuro-oncology. Despite decades of research in this field, no consensus has emerged as to the best approach to tackle this physiological limitation. Moreover, the relevance of doing so is still sometimes questioned in the community. In this paper, we present our experience with CNS delivery strategies that have been developed in the laboratory and have made their way to the clinic in a continuum of translational research. Using the intra-arterial (IA) route as an avenue to deliver chemotherapeutics in the treatment of brain tumors, complemented by an osmotic breach of the blood-brain barrier (BBB) in specific situations, we have developed over the years a comprehensive research effort on this specialized topic. Looking at pre-clinical work supporting the rationale for this approach, and presenting results discussing the safety of the strategy, as well as results obtained in the treatment of malignant gliomas and primary CNS lymphomas, this paper intends to comprehensively summarize our work in this field.

## 1. Introduction

Chemotherapeutic drug trials for brain tumor treatment have been conducted worldwide for many decades, with marginal improvements in patient outcomes. Indeed, the standard of care in the 1st-line management of glioblastoma (grade 4 primary brain tumors) was the addition of temozolomide, an alkylating drug, to radiotherapy, which led to an improvement in survival of 2 months [1]. This regimen is dubbed the “Stupp regimen”. Any further attempts to improve on the outcome have produced disappointing results. Interestingly, one of the only reported approaches with seemingly improved outcomes is the addition of a local device emitting low-intensity, intermediate-frequency alternating electric fields (TTF) [2]. As this device is applied directly to the scalp, and its effect does not require a specific delivery paradigm to reach the CNS. Indeed, amongst the factors that can explain a lack of improvement in the care of brain tumor patients, one stands as a major culprit: Impaired delivery to the CNS, related to the presence of the blood-brain barrier (BBB) [3]. Thus, in the presence of a brain tumor, the first barrier to treatment options is just that, a barrier: The BBB. 

## 2. The Blood-Brain Barrier

It has been a long process to recognize the extent to which the BBB really impacts CNS delivery. It is often still debated in some publications, as some authors continue to argue that the presence of contrast enhancement on computed tomography(CT) scans (iodine-based) or on magnetic resonance scans (para-magnetic contrast) remains clear evidence that the integrity of the BBB is altered and access to the CNS is granted [4,5]. Hence, in that context, these authors claim that the BBB entity does not represent a significant obstacle to therapeutic delivery to the CNS in the presence of pathological lesions, implying that the breach in permeability is sufficient to allow adequate diffusion of therapeutics. This type of all or none argumentation basically translates a lack of knowledge and understanding of BBB alterations and CNS delivery subtleties in the presence of a tumor. Indeed, different pharmacokinetic compartments are defined by the presence of a brain tumor, with a wide variation of the effects on the BBB permeability, and thus, on delivery [6]. This aspect is frequently neglected and under-estimated.

Indeed, looking at modern data on the subject, there is no doubt that the BBB prevents chemotherapy entry to the CNS, even in the presence of a lesion, thereby limiting therapeutic concentration from reaching clinically efficient levels [7,8]. Part of the confusion arises from the fact that within a brain tumor, as well as to the close proximity of the tumor nodule, the BBB is often replaced by a brain-tumor barrier (BTB) which pertains to entirely different pharmacokinetics, displaying a permeability that is classically intermediate between normal BBB and breached BBB. This increase in permeability is a function of the breach in the integrity of the BBB and BTB, and is highly variable, heterogeneous, and dependent on tumor size and type [9,10,11]. Thus, within any tumor, drug distribution is inherently uneven, with preferential accumulation in the necrotic central core areas [12], whereas drug penetration at the edge of the tumor is classically nonexistent, or marginal at best [13]. As such, although the BBB and the BTB are often partially breached, there remains a significant delivery impediment, and therapeutic levels of drugs are insufficient within the breached areas to mount a clinically significant response [6,14]. As was eloquently exposed by Reichel, despite enormous efforts, achieving an effective concentration profile in the brain remains a significant challenge in CNS drug development [6]. Another factor further complicates the matter: The majority of malignant brain tumors are infiltrative, with tumor cells permeating at a distance from the main tumor nodule, and away from even the most sensitive imaging MR scan sequence (FLAIR) [14,15]. Obviously in these areas, the BBB permeability is unaltered, and tumor cells are shielded by an intact BBB.

## 3. Alternate Drug Delivery

Different approaches have been designed and tested to circumvent this delivery impediment, bypass the BBB and BTB, and maximize therapeutics delivery to the brain. Indeed, in order to increase the number of therapeutic options available to treat CNS tumors, alternative delivery strategies have to be considered. For a detailed review on the subject covering different strategies, please consult the review by Drapeau et al. [3]. Of all the approaches we have tested in the laboratory, one is currently used in the clinic: The cerebral intra-arterial infusion of chemotherapy (CIAC) with and without BBB permeabilization. This paper will give a thorough description of the efforts carried by our group to successfully deploy this strategy in the treatment of brain tumors. Indeed, we have implemented a continuum of translational research on this topic that will be described in detail in this publication.

### The Cerebral Intra-arterial Infusion of Chemotherapy (CIAC) and the Blood Brain Barrier Disruption (BBBD) Adjunct

When one realizes the extensiveness of the vascular network supplying the brain, it becomes obvious that a global delivery strategy is rational and plausible by using this vascular network as a delivery corridor [16]. The importance of this network has already been exposed by Bradbury and colleagues, these authors claiming that the entire network covers an area of 12 m^2^/g of cerebral parenchyma [17]. To understand the extensiveness of the cerebral vascularization in a more prosaic statement, let us just consider that the brain receives 20% of the total systemic circulation even though its weight amounts to less than 3% of the total body weight [9].

Interestingly, it is technically easy and actually commonly performed in the clinic to repeatedly access this cerebral vascular network in a patient [18]. Via a simple puncture to access the femoral artery, a catheter is introduced and navigated intra-arterially to reach one of the four major cerebral arteries. Once in position, the chemotherapy is administered via the catheter that is withdrawn at the conclusion of the procedure. The CIAC produces a paradigm of regional chemotherapy distribution within the area deserved by the vessel treated [19].

Through the first pass effect, an increase in the local plasma peak concentration of the drug produces a significantly improved AUC (the concentration of the drug according to the time) [19,20]. This consequently translates in increased local exposure of the target tissue to the therapeutic agent. Interestingly, as our lab has shown, it is also accompanied by a decreased systemic drug distribution, hence reducing systemic toxicities and potential side-effects [21]. Classically, the therapeutic concentration at the tumor cell target is increased by a 3.5–5-fold factor [20]. This procedure is performed under local anesthesia, and typically lasts around 30 min.

The delivery can be further improved by adding as an adjunct to the procedure an osmotic blood-brain barrier disruption (BBBD). This strategy is based on the cerebral intravascular infusion of a hypertonic solution to produce a transient increase in permeabilization of the BBB and BTB, prior to the administration of the chemotherapy. As with CIAC, the parent vessel treated is selected based on the tumor localization in the brain. This approach, which is an adjunct to the CIAC, is physiologically more demanding, requiring general anesthesia, and needs careful preparation, but it does increase significantly delivery across the BBB and BTB [3,9]. It involves the IA infusion of a hyperosmolar solution (usually mannitol) in a flow rate sufficient to allow a complete filling of the vessel. Two parameters are paramount in the ability to mediate a hyperosmolar modification of the barrier: The osmolality of the solution, and the infusion time. Using a solution of 1.6 molal arabinose in pentobarbital-anesthetized rats, Rapoport determined an interval duration of 30 s as the optimal infusion time to produce a BBBD [9]. The same infusion time was applied to the use of mannitol with similar findings in the same animal model [9]. These parameters have made their way to the clinic, albeit the anesthetic agents are now different.

The combination of IA infusion of a molecule with osmotic BBBD has been shown to further increase the effect of the first pass through the brain, increasing maximal peak concentration as well as AUC of the administered molecule [16,22,23]. Sato et al. elegantly presented in vivo data showing that BBBD produces a marked increase in permeability at the edge of the tumor. Interestingly, this area is typically associated with active tumor cells proliferation, whereas the permeability of the BBB and BTB tends to renormalize [13]. Theoretically, the concept of beaching the permeability of the BBB is quite compelling, as it could help evade the “sink effect” by providing higher and more uniform delivery to a whole CNS vascular territory, allowing prolonged tumor cell exposure to higher concentrations of the administered therapeutics [6,16]. This sink effect is triggered by areas of necrosis within the tumor, which tends to attract and concentrate the chemotherapy crossing the CNS, stealing the peripheral areas of the tumor where the drug would be most useful [16]. Obviously, this includes the neoplastic cells at the tumor edges that are often the most proliferative and protected by an intact BBB and/or BTB [14,15,16,24].

## 4. Pre-clinical Data 

While numerous investigators have studied CIAC and BBBD over the years, we undertook a thorough pre-clinical characterization in the Fischer-F98 model to ascertain, objectivate, and measure the delivery advantage provided by both approaches. We first characterized the F98-Fischer glioma model as a benchmark for our delivery studies. The model was found to be highly predictive and reproducible in term of tumor growth dynamics and animal survival (Figure 1). 

Using a standardized implantation procedure, the tumor-take has systematically been 100%, with a median survival of 26 ± 2 days [25,26]. Figure 1A shows the position of the animal in the stereotactic frame for precise insertion of the needle in the brain of the animal using a precise and standardized coordinate system [25]. This is paramount for reproducibility across experiments. Indeed, a free-hand implantation technique which is frequently employed in the literature is inadequate for these types of studies. Likewise, we found that the use of a micro-infusion pump is essential in minimizing tissue damage and associated inflammatory reaction triggered by the implantation process [25]. The slow (1 μL/min) and steady infusion rate and the low volume (10,000 cells in 5 μL) ensures minimized cerebral tissue disruption and prevents the backflow along the implantation track commonly associated with these models [25,26]. This produces a constant pattern of tumor growth in the right hemisphere of the animal, where the tumor is already noticeable at day 3 post-implantation (Figure 1B), and starts to produce an alteration in consciousness around day 26. Experimental treatments are performed at day 10 post-implantation, when the tumor has reached a significant size (Figure 1B), without altering the neurological functions of the animal [27,28,29,30].

Using this model, and based on slight alterations of the methods described by Neuwelt and his team [16], we developed a technique allowing the perfusion of therapeutics via the intra-arterial (IA) route in the carotid of the Fischer rats, while under general anesthesia in an MR gantry. This allowed us to study the dynamics of real-time imaging during the infusion of any selected MR traceable molecule (Figure 2) [22]. 

In this particular surgical montage, the right external carotid artery has been identified, incised, and cannulated using a PE50 catheter. Once in position, any solution can be perfused in a retrograde fashion via the external carotid artery into the internal carotid artery (Figure 2). When high flow solutions are infused, such as when we perform a BBBD, a clip is secured on the common carotid artery to isolate the system from the heart, and prevent downstream backflow of mannitol to the heart. Once terminated, the clip is removed, the external carotid artery is simply ligated, and the incision is closed.

As an initial experiment, we first characterized the baseline level of CNS entry for 2 paramagnetic compounds, Magnevist (743 Da) and Gadomer (17,000 Da) in tumor-bearing F98-Fischer rats. As expected, the smaller Magnevist displayed a greater than 3-fold baseline penetration in the tumor compared to Gadomer across the BTB, whereas penetration in the BBB around the tumor was negligible, and was no different than in the contralateral hemisphere for both molecules [22,23]. 

Next, we studied the concentrations of different platinum drugs when administered via different routes: Intravenous (IV), IA, and IA + BBBD using inductively coupled plasma mass spectrometry (ICP-MS) in the Fischer-F98 rat model. We did so for 5 platinum: Cisplatin, Carboplatin, Oxaliplatin, Lipoplatin and Lipoxal [28]. Figure 3 shows the summary of these experiments. Ten days after the F98 glioma cells implantation, the platinum drugs were administered according to the selected route of administration. Equivalent doses of platinum to those used in humans were established based on the body surface area of the animals [28]. Animals were euthanized 24 h after the drug perfusion, brains were harvested, and cut in sections with a brain matrix [27]. The tumor was separated and divided into cytoplasmic and nuclear compartments using a commercial Nuclear Extract Kit (Active Motif, Carlsbad, CA, USA) for analysis by ICP-MS [28].

Looking specifically at the concentration of platinum reaching the nucleus of the tumor cells, we observed significant differences between the different routes of administration (Figure 3). Comparing IA against IV, an increase in the order of 20-fold was observed for IA Carboplatin, whereas it reached 40-fold for Lipoplatin and 90-fold for Lipoxal! Interestingly, these studies also depicted significant neurotoxicity when experimenting IA infusion of either Oxaliplatin or Cisplatin, hinting at the fact that these 2 drugs were not suitable candidates for IA delivery [27,28,29]. These increases in the tumor cells nuclei delivery were even more dramatic when a BBBD was added to the IA infusion. Specifically looking at Carboplatin, the IA + BBBD further increased the delivery by a 17-fold factor compared to IA alone, a 320-fold factor compared to the IV infusion [29].

Using the same experimental design, we also assessed the delivery of Temozolomide. Temozolomide is the first-line standard of care in the treatment of primary brain tumors. As the bio-disponibility of the oral formulation is close to 100%, the IV formulation is available but rarely used in the clinic. In the present study, the IV formulation was used to emulate clinical oral administration. Hence, we tested the delivery of IA, IA + BBBD, and IV Temozolomide in the Fischer-F98 glioma model. The animals were once again treated 10 days after implantation. Using liquid chromatography with tandem mass spectrometry (LC-MS/MS), we measured Temozolomide in plasma, CSF and brain at 3 timepoints post-Temozolomide infusion [21]. Compared to IV, we found a fourfold increase in Temozolomide peak concentrations in brain tumor tissues with IA infusion, and a 5-fold increase with BBBD [21] (Table 1 and Table 2). 

The increase was not as dramatic using the BBBD as an adjunct with IA of Temozolomide, compared to the platinum compounds. The values of c max according to the route of delivery were as follows: 10.582 (IV), 42.989 (IA), and 50.751 (IA + BBBD), respectively. In this paper, although we could measure a significant increase in Temozolomide delivery as described above, we did not observe a parallel increase in survival of the treated animals. In vitro characterization of the F-98 glioma cell line showed it to be resistant to temozolomide [21]. Hence, it is obvious that delivery is not the only factor at play, as will be discussed later.

These pre-clinical results really highlight the potency of IA and IA + BBBD as an adequate route of delivery to improve the different pharmacokinetic parameters of CNS therapeutic delivery. The pre-clinical research continuum to improve and maximize these procedures continues, as each therapeutic offered by this route first needs to be tested for innocuity in animal models to rule out any major toxicities. Indeed, Taxol, cisplatin, and oxaliplatin were found to be extremely toxic in pre-clinical testing, excluding these drugs as eventual candidates for IA delivery. Moreover, as can be derived from the results obtained with the temozolomide experiments, an increase in delivery is not necessarily associated with an improvement in outcome. Hence, delivery is only one of many aspects of therapeutic success in the treatment of CNS tumors, albeit an important one. We will further discuss this issue in the next section on the clinical applications of these procedures.

## 5. Clinical Procedures

The access to the arterial system is obviously accomplished differently in humans. The human cerebral arterial system is organized in such a way that there basically are 4 major arteries responsible for the brain irrigation (2 carotids, and 2 vertebral arteries). The vascular anatomy can be variable from one individual to another, and thus the precise anatomy must be determined during the first treatment session by a formal cerebral angiography. If a lesion covers more than one vascular distribution, or if there are multiple lesions, the treatment is delivered by equally splitting the chemotherapy dose in the different distributions (vessels) involved. Parameters such as catheter placement, dilution, and rate of infusion are all standardized. In the human, the arterial system is accessed via a percutaneous transfemoral puncture. Once accessed, the catheter is navigated in the arterial system using radiological imaging (fluoroscopy). As shown in Figure 4A, the catheter has been placed in the left carotid artery, and a contrast infusion shows the distribution of this vessel. 

The technique involves the following steps:

1. Selective catheterization is performed via percutaneous transfemoral puncture of the left internal carotid artery, right internal carotid artery, left vertebral artery or right vertebral artery. The tip of the catheter is positioned at the C2-C3 vertebral level in the carotid (Figure 4), or at the C6-C7 vertebral level in the vertebral artery.

2. Infusion of the drug IA: When infusing intra-arterial solutions, the concentration of the solution and the rate of infusion are critical factors that need consideration in avoiding neurotoxicity. The phenomenon of streaming defines an inhomogeneous distribution of the administered solution because of poor mixing at the infusion site [9]. Density and viscosity of fluid, lumen diameter of the infused vessels, and velocity of flow are all important determinants to control in order to avoid streaming. In this case, the Caelyx, Melphalan, and Etoposide phosphate are infused at a rate of 0.12 cc/s, whereas the Carboplatin and Methotrexate are infused at a standard rate of 0.2 cc/s.

3. In the case of a BBBD: BBBD procedures require general anesthesia. Hence, after general anesthesia with Propofol, we proceed to a selective catheterization via percutaneous transfemoral puncture of the treated artery. We then determine the individual rate of infusion of Mannitol. We use iodinated contrast injection and fluoroscopy to establish, for each patient, the ideal infusion rate; it is the rate that will fill the entire vessel distribution, without producing significant reflux in the common carotid artery. Once established (usually between 3 and 6 cc/s × 30 s), the patient is prepared for the hemodynamic repercussions of the procedure. Indeed, the osmotic disruption is a physiologically stressful procedure. It can induce focal seizures in 5% of procedures. It can also trigger a vaso-vagal response with bradycardia and hypotension. In order to prevent the occurrence of these adverse effects, the following medications are administered just prior to the disruption: Diazepam 0.2 mg/kg IV (maximum dose = 10 mg), and Atropine IV, titrated to increase heart rate 10–20% from baseline (0.5–1 mg). We then proceed to the BBBD, after which IA infusion of chemotherapy is accomplished. Figure 4B shows the repercussion of BBBD on delivery. In this image, an IV contrast material was infused shortly after the BBBD (within 5 min), showing a diffuse penetration of the contrast compound in the brain parenchyma (arrow).

### 5.1. CIAC or CIAC + BBBD? A Question of Intensity of Delivery

The question of whether to use CIAC alone or with an adjunctive BBBD really is a question of intensity in the amount of delivery. There is no question that BBBD will increase delivery compared to an isolated CIAC. When studying platinum compounds, this increase has been shown to be variable for each molecule, providing a 2-fold increase for Carboplatin (overall), and up to a 5-fold increase for Lipoxal compared to IA alone (Figure 3). However, the use of BBBD requires general anesthesia, and is significantly heavier for the patient. Hence, its use can be limited by the availability of anesthesia and all it implies (recovery room, etc.). On the other hand, CIAC is easy to perform, and virtually devoid of these limitations. The procedure is cheap, and the only limitation is the access to the angiography suit. Hence, we have traditionally reserved the use of BBBD for patients with potentially curable diseases, such as primary CNS lymphomas (PCNSL). Metastatic brain disease, as well as glial tumors, are typically treated by CIAC. We built most of our clinical studies around a model in which the patient receives a monthly treatment session, typically up to 12 sessions. Only in patients of these 2 groups of pathologies presenting a complete response or near-complete response will we consider using BBBD to consolidate the treatment response in the last 2 cycles of treatment. 

### 5.2. Clinical Data: Safety

Neurotoxicity is a legitimate concern when deploying a strategy that increases CNS delivery of therapeutics. Indeed, transgressing the BBB could result in an increase in neurotoxicity. Thereby each therapeutic used in the clinic has been previously tested in the animal model to screen for compatible drug candidates for CIAC/CIAC + BBBD clinical use. Obviously, this does not entirely preclude the risks of toxicity. However, now looking at the modern series of CIAC/CIAC + BBBD, we can confidentially claim it to be safe, when performed in expert centers. 

Doolittle et al. reported on the experience of the BBBD consortium, a multi-site consortium performing CIAC with and without BBBD for malignant brain tumors [30]. These authors concluded that with standardized protocols, CIAC was safe across multiple centers, with a low incidence of catheter-related complications. In their series of 221 patients treated between 1994 and 1997, they observed a sub-intimal tear rate of 5%, whereas the rate of strokes was 1.7%. 

We undertook a detailed review of our own experience in terms of complications, going into further details. We analyzed our entire cohort of CIAC patients to brush the best possible picture in terms of innocuity. Between January 2000 and June 2015, a total of 3583 arteriographic procedures for CIAC/CIAC + BBBD were performed on 722 patients in the treatment of brain tumors at CHUS (centre hospitalier universitaire de Sherbrooke, Sherbrooke, Canada). All patients were afflicted by a malignant brain tumor (463 primary brain tumor, 158 metastasis, 101 lymphomas). To our knowledge, this is the largest such series available in the literature [31]. 

As clinical data have been cumulated prospectively in the context of clinical studies, data were extracted from all hospitalization records for care events related to a CIAC procedure in the treatment of brain tumors (glial tumors, PCNSL, and metastatic tumors). Complications were studied and grouped under 3 different headings: Vascular complications, per-procedural epileptic manifestations, and hematological toxicities. The results are detailed in Table 3.

#### 5.2.1. Vascular Complications

Overall, a total of 66 vascular angiographic or MRI incidents were uncovered (1.84%). More specifically, 5 asymptomatic dissections were observed, 9 asymptomatic carotid stenosis and 3 occlusions were identified, 2 which were symptomatic. (Table 3). 

In terms of cerebral newly described lesions, the MRI identified 5 acute hemorrhagic strokes (1 symptomatic), 38 lacunar strokes (20 symptomatic), and 6 acute ischemic lesions (4 symptomatic). One of these strokes in the posterior fossa was a catastrophic event that led to the patient’s death. Overall in this series, the total number of symptomatic vascular complication rate was 27 (0.75%).

#### 5.2.2. Seizure Events

The overall per-procedural seizure incidence was 2% (74 incidents) as can be appreciated in Table 3. Of these, 9 were generalized seizures, whereas 65 were partial seizures. Interestingly, a simple discontinuation of the chemotherapy infusion was sufficient to halt the seizure in all patients, but one. Most seizure fits (84%) were observed during Methotrexate infusion for primary CNS lymphomas (PCNSL). Only 3 partial seizures were observed in the treatment of glial tumors.

#### 5.2.3. Hematological Complications

Hematological complications were classified according to the National Cancer Institute of Canada (NCIC) toxicity criteria. A total of 11.4% grade 3 and 7.2% grade 4 toxicities were observed.

Hence, from the analysis of this data, we feel justified to conclude that the procedure is safe, and its use is appropriate in this clinical context. This affirmation does imply that the treatments are performed in expert centers, and the therapeutics used with CIAC have been screened with an adequate methodology and are known to be devoid of neurotoxicity. While the technical details of the procedure are beyond the scope of this publication, a few considerations need to be discussed. First, although some might argue that a supra-selective catheter placement might be of interest, we always use a proximal position in the treated vessel (C1-2 for the carotid, C2-C3 for the vertebral). The rationale supporting this has to do with the infiltrative nature of most brain tumors, always lending for a more widespread disease than the MR scans actually reveal. This is true for glial tumors, PCNSL, but also metastasis. Because of that, we see little interest to target for a supra-selective catheterization, especially considering that this approach would likely increase the risks of complications while limiting the actual distribution of chemotherapy to the CNS.

Secondly, each therapeutic comes with its own set of infusion parameters based on the concentration, density, and volume of the infusion solution. This is paramount to minimize the risks of neurotoxicity related to concentration and streaming.

## 6. Clinical Results 

We will focus the discussion on the results obtained for GBM (grade 4 astrocytomas) and PCNSL. As a remainder, all patients with GBM were treated by CIAC, whereas 10% of these receive CIAC + BBBD as a consolidation procedure for the last 2 cycles. PCNSL patients were all treated with CIAC + BBBD.

### 6.1. Glioblastoma (GBM)

In our series, 319 GBM patients were treated by CIAC. Treatment sessions were performed every 4 weeks, unless hematologic parameters prevented it. Carboplatin is the drug of choice, either alone, or combined with either Melphalan or Etoposide phosphate, depending on the protocol used. All GBM patients were treated at relapse. Seventeen percent were treated at first relapse, 68% at second relapse, 11% at third relapse and 4% at fourth (Figure 5). 

The fact that most of our patients were treated at 2nd relapse unfortunately negatively biases our results. 

Overall, patients received a median of 4 cycles (1–22 cycles). Progression-free survival was 5 months. The whole series presented an overall median survival of 25 months, and survival from study entry was 8 months for the entire cohort (Figure 6). This is superior to most commonly reported treatments at relapse, which usually produces median survival from treatment initiation of 4–6 months [2,3,7].

Ten patients are still alive, with the longest survival now at 16 years. When looking exclusively at those patients treated at 2nd relapse, median survival jumps at 11 months from the study entry for the entire cohort. 

Looking at the best radiological responses obtained according to the RANO criteria, we found the following: 23% of patients have shown progression, 26% have presented a stabilization of their disease, 42% have shown a partial response (Figure 7), and 6% a complete response.

Although these results are encouraging, comparing this data with modern series is complicated by 2 factors. Our data on GBM is plagued by a major weakness: Heterogeneity. Indeed, in 2016, the classification on gliomas changed, and one major overhaul has been the inclusion of molecular markers to better stratify the patients in prognostic classes that strongly determine their evolution [32]. The IDH marker is the stronger determinant of survival, and, nowadays, most modern series stratify patients according to this marker, which we did not. The other source of heterogeneity in this series pertains to the fact that a majority of patients were exposed to multiple treatments prior to accrual (Figure 5). These 2 factors are obviously now considered in the design of our studies. Recruitment and stratification is now refined to eliminate these confounding factors. 

Ideally, to avoid this heterogeneity, a randomization process should also be utilized. We tried to launch such a study a few years back, but were faced with difficulty as to what should be the randomized arm. The design of the randomized study is now complete and submitted. It will compare carboplatin/Etoposide phosphate CIAC against oral Lomustine (CCNU) in the control arm, and will constrain accrual at 1st relapse, as well as stratify patients against molecular status. Hopefully, this will allow us to demonstrate the superiority of this approach, once and for all.

### 6.2. Future Perspectives in the Treatment of Malignant Gliomas

#### 6.2.1. Heterogeneity of Response: The Impossibility to Predict the Best Regimen for Each Patient

Carboplatin is our drug of choice, as it appears to be the most effective, producing responses in 70% of patients for a median PFS of 5 months at relapse. However, we do have an array of agents available for intra-arterial infusion in the clinic, and we continue to expand this list: Carboplatin, Methotrexate, Melphalan, Etoposide phosphate, and more recently, Caelyx. These agents have all been used safely in CIAC for brain tumor treatment by our team. Interestingly, in the case of non-response to Carboplatin, other agents can be used in subsequent cycles of treatment for patients still presenting an adequate functional status. In these cases, we are often confronted with extremely variable results, with some long-term responses (up to 180 months) observed with other agents than Carboplatin, whereas some patients show no response whatsoever to any agents. Indeed, these tumors all appear to have their own distinctive sensitivity profile to chemotherapy agents, and we believe that they should therefore all be approached as a singular disease entity requiring a personalized treatment. Molecular stratification has come a long way in the management of glial tumors [32], but its role is limited in assisting pathological stratification and prognosis. It is not yet used in treatment selection. We propose to combine data from in vitro drug sensitivity testing (DST) and molecular characterization using “The Cancer Genome Atlas” (TCGA) stratification, in addition to a panel of chemoresistance markers, to select the best drug candidates prior to the initiation of CIAC. Hence, in accordance with this scheme in a proposed clinical study at 1st relapse for GBM, all patients will be re-operated prior to the beginning of CIAC. During surgery, a tumor sample will be obtained for the DST, molecular stratification and chemoresistance panel markers, and the treatment will be tailored specifically to each patient.

#### 6.2.2. Radio-Chemotherapy

Another area we have started to explore is the combination of Carboplatin with radiation therapy [33,34]. Indeed, radiotherapy is the most effective single-treatment modality for GBM tumors, but it controls the disease only transiently. A way to improve treatment consists of coupling radiation with a potent radiosensitizer. Carboplatin, a platinum (Pt) drug, is ideally suited for this. Our group has demonstrated that the addition of Carboplatin to ionizing radiation produced significantly more DNA strand breaks [35,36,37,38,39]. In numerous cell lines, combining radiotherapy and Carboplatin was found to increase cell death. In a mouse model, we observed a maximum antitumor effect with Carboplatin administration at 4 or 48 h prior to irradiation. This timing correlated to the highest levels of Pt bound to DNA [35,37,39]. Concurrent Carboplatin and radiation treatment represent a common modality for treating a variety of cancers. Unfortunately, since this class of drug does not readily cross the BBB when administered via the standard IV routes, they are not used to treat GBM. We have just started accrual on a new phase II study in which we administer IA Carboplatin with a re-irradiation protocol in a dose escalation scheme. We feel that this combination has the potential to improve clinical results. We have 6 patients (of a total of 35) recruited, and enrollment is ongoing.

### 6.3. Primary CNS Lymphomas

PCNSL are a rare and aggressive form of central nervous system tumors. Generally confined to the brain, eyes and/or cerebrospinal fluid compartments, these extra nodal non-Hodgkin large B cell lymphomas typically show no evidence of systemic diffusion [40]. PCNSL is an extremely aggressive disease, with a median survival time of 3 months without treatment [40]. It is a fairly unusual occurrence, accounting for 1% of cases of lymphoma, whereas it represents 4% of primary brain tumors [41,42]. A current trend in the treatment of this disease has been radiation therapy avoidance, as it was shown to be extremely neurotoxic to patients [43]. Over the years, different protocols of IV high-dose Methotrexate have shown encouraging results. Indeed, Da Broi et al. reported the results of 57 patients treated over 12 years with chemotherapy [44]. Overall, they found a median OS of 35.4 months, and a PFS of 15.7 months. Using CIAC + BBBD infusion protocol of high-dose Methotrexate (combined to Etoposide, Cyclophosphamide and/or Procarbazine), Angelov et al. reported a median overall survival of 3.1 years. They also reported the neuropsychological outcome profile in 26 long term survivors from the treatment (median follow up of 12 years), showing the innocuity of this approach [45]. This good quality data shows without a doubt that repeated CIAC + BBBD infusion protocol of high-dose Methotrexate does not impact the neurocognitive functioning of responding patients.

Using CIAC + BBBD Carboplatin (400 mg/m^2^) in addition to high dose IA Methotrexate (5 g), we treated 43 newly diagnosed PCNSL patients from 1999–2018. The median age of the cohort was 63, with a mean age of 60 years old. The cohort was comprised of 24 males and 19 females. Overall, remission was induced in 34 patients (79%). The overall median survival was 46.5 months for the entire cohort. Actuarial survival was 88%, 64%, 54%, 39% and 18% at 1, 2, 3, 5 and 10 years. The progression-free survival for the entire cohort was 43.3 months. The actuarial PFS was 83%, 59%, 56%, 30%, and 9% at 1, 2, 3, 5, and 10 years. These are amongst the best results ever published in the treatment of this disease, without the use of radiation therapy. The detailed manuscript presenting these results is in preparation.

## 7. Conclusions

Intra-arterial chemotherapy is a delivery vehicle allowing the increase of available therapeutics in the treatment of brain cancers. Its initial use many decades ago has been hampered by toxicity, a problem which is no more of concern. Angiographic refinements, combined with intra-arerial infusion of therapeutics carefully selected for this purpose have rendered this approach safe and sound. The addition of an osmotic permeation of the BBB further increases delivery of therapeutics to the CNS. We need to acknowledge the extreme heterogeneity of GBM, and eventually start tailoring treatment to each tumor for individual patients in order to improve the modest results obtained so far. Drug selection is at the core of this process. We also need to keep expanding the pool of agents that can safely be administered via this route. As for the treatment of PCNSL, different refinements are considered to keep improving outcomes in the treatment of this disease. The addition of rituximab, a CD-20 antibody, should be considered as an adjunct to the treatment protocol. In the end, the use of intra-arterial therapeutics infusion combined to osmotic blood-brain barrier permeation answers the need to adequately address an issue that is commonly underestimated: The presence of the BBB, and the complex pharmacokinetic set of compartments it imposes on CNS delivery. 

## Figures and Tables

**Figure 1 pharmaceutics-11-00248-f001:**
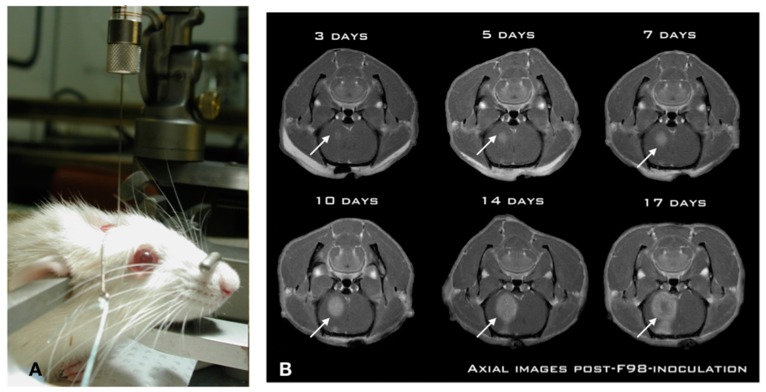
The Fischer-F98 glioma model shows a reproducible and predictable growth pattern. (**A**) The infusion of the cell suspension is accomplished using a slow steady perfusion with a micro-infusion pump. Also, 10,000 cells are infused at a rate of 1 μL/min over 5 min. (**B**) Coronal views of an implanted animal at days 3, 5, 7, 10, 14 and 17 post-implantation. Notice the gradual progression of the gadolinium enhancement on the MR scans in the right hemisphere (arrows) depicting the steady tumor progression. The animal starts to develop faint subtle symptoms (lateralization) at day 14, that culminate at day 26 ± 2 days.

**Figure 2 pharmaceutics-11-00248-f002:**
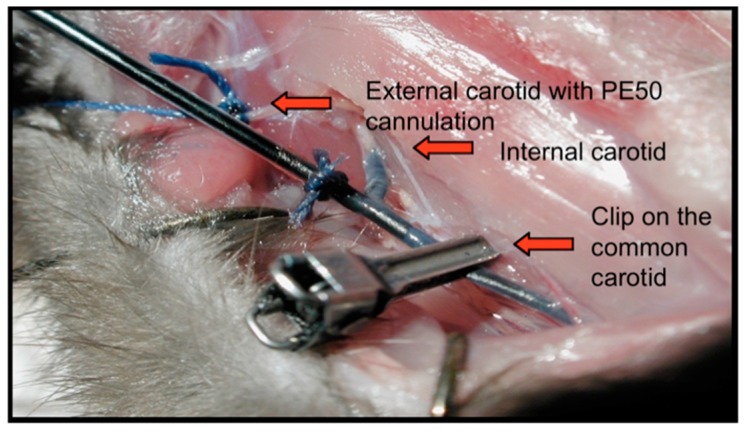
The surgical montage for intraarterial infusion and blood-brain barrier disruption (BBBD) in the Fischer rat. Once the montage is ready, the animal can then be inserted in the magnetic resonance (MR) gantry for real-time imaging. The intraarterial carotid perfusion is accomplished in a retrograde fashion via the external carotid artery. As can be appreciated on this image, a clip is also placed on the common carotid artery to prevent downstream backflow. As soon as the infusion is completed, the clip is removed, and the external carotid artery is sutured.

**Figure 3 pharmaceutics-11-00248-f003:**
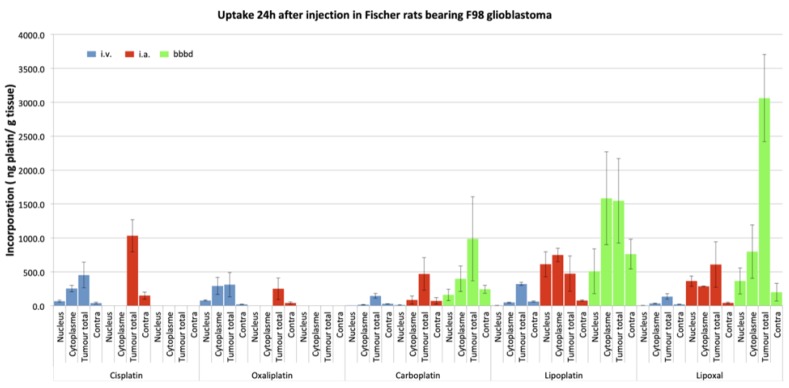
A comparison of the 5 platinum drug accumulations as related to the route of infusion, measured by inductively coupled plasma mass spectrometry (ICP-MS) in the Fischer-F98 rat. As can be observed, the intra-arterial (IA) and IA + BBBD routes were not tested for Oxaliplatin and Cisplatin because of significant toxicity. Results are reported as measurements of platinum (ng pt/g tissue) in the nucleus, cytoplasm, and whole tumor. The magnitude of the increase observed for each platinum agent can be appreciated in relation to the route of delivery. There is a significant increase in platinum delivery (ng platinum/g tissue) with all molecules, except Oxaliplatin.

**Figure 4 pharmaceutics-11-00248-f004:**
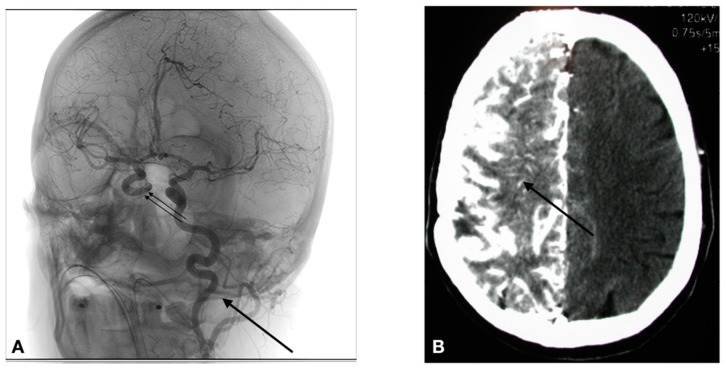
(**A**) Catheter placement in a glioblastoma patient treated for BBBD in the left carotid artery (arrow). An iodine contrast was infused, opacifying the left carotid distribution, as well as the contralateral carotid (double arrow) via the polygon of Willis. (**B**) The image produced by a BBBD of the right carotid artery on a computed tomography (CT) scan in a patient afflicted by a primary central nervous system (CNS) lymphoma after an infusion of iv iodine contrast. As can be appreciated, the whole hemisphere is bathed by the contrast, an evidence by the fact that the BBB is breached.

**Figure 5 pharmaceutics-11-00248-f005:**
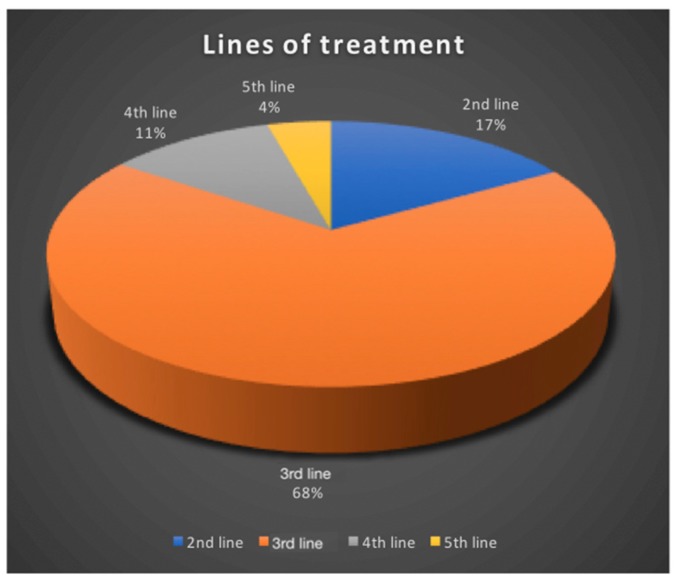
A breakdown of the number of treatment lines to which GBM patients were exposed prior to accrual in our series. As can be appreciated, most patients were exposed to 2 lines of treatment (68%) prior to accrual.

**Figure 6 pharmaceutics-11-00248-f006:**
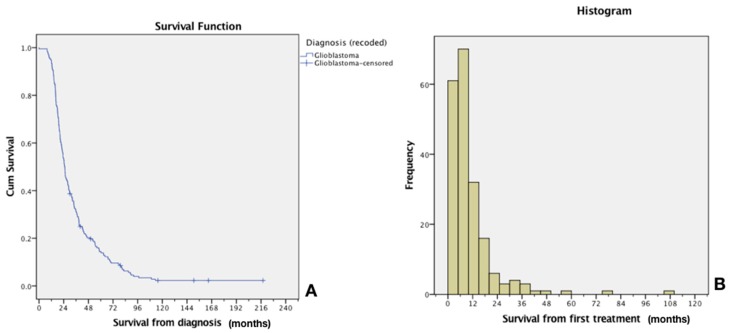
Median survival from diagnosis (**A**) and distribution survival histogram (**B**) of GBM patients exposed to CIAC. Median survival from diagnosis was 25 months, whereas it was 8 months from the study entry. As can be appraised from the distribution histogram, most patients progress and die from their disease in the first 12 months after accrual. This leaves around 25% of patients whose survival is greater than 12 months.

**Figure 7 pharmaceutics-11-00248-f007:**
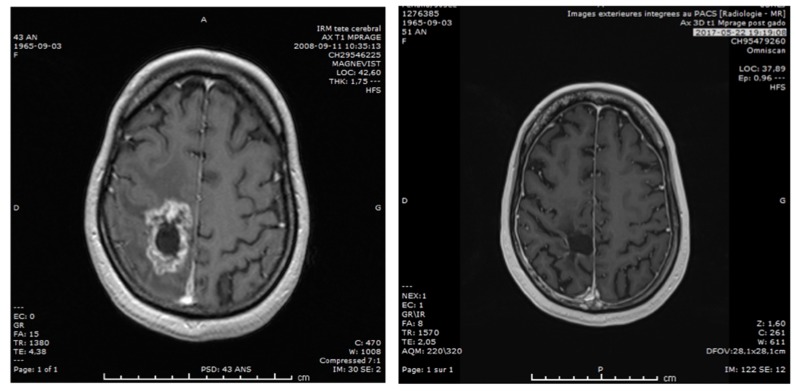
Example of one of our best responder: A 43-year-old female glioblastome (GBM) patient treated with intraarterial Carboplatin/melphalan in 2008, who remains in complete response in 2017. This patient was treated at first relapse, without using any other therapy than the Stupp regimen at the first-line.

**Table 1 pharmaceutics-11-00248-t001:** IV administration of Temozolomide (TMZ) (200 mg/m^2^).

CNS Compartment Analyzed	*T*_1/2_ (h)	*T*_max_ (h)	*C* _max_	*AUC* _0–*t*_
**Plasma**	1.08	0.25	63,581 μg/mL	53,409 h·μg/mL
**CSF**	0.87	0.25	7,628 μg/mL	6,658 h·μg/mL
**Tumor**	1.51	0.25	10,582 μg/g	9,521 h·μg/g
**Lpsilateral brain**	0.66	0.25	10,273 μg/g	8,530 h·μg/g
**Contralateral brain**	0.83	0.25	9,790 μg/g	8,547 h·μg/g

**Table 2 pharmaceutics-11-00248-t002:** IA administration of TMZ (200 mg/m^2^).

CNS Compartment Analyzed	*T*_1/2_ (h)	*T*_max_ (h)	*C* _max_	*AUC* _0–*t*_
**Plasma**	n/a	0.5	40,676 μg/mL	38,759 h·μg/mL
**CSF**	1.89	0.25	8,436 μg/mL	7,681 h·μg/mL
**Tumor**	0.34	0.25	42,989 μg/g	31,934 h·μg/g
**Lpsilateral brain**	0.35	0.25	31,056 μg/g	23,930 h·μg/g
**Contralateral brain**	n/a	0.5	11,714 μg/g	10,130 h·μg/g

Temozolomide (TMZ) pharamacokinetic parameters measured by Liquid chromatography tandem-mass spectrometry (LC-MS/MS) in Fischer-F98 rats treated 10 days after tumor implantation. Parameters are compared between the IV route (Table 1) and the IA route (Table 2).

**Table 3 pharmaceutics-11-00248-t003:** Angiographic, seizure-related and hematologic complications in the series of CIAC/CIAC + BBBD patients treated in Sherbrooke, from 2000–2015. A total of 3583 procedures in 722 patients were accomplished.

Angiographic + Vascular Complications	Number of Event MRI + Angiographic Findings	MRI Findings	Symptomatic Lesions (the Lesion was Accompanied by Clinical Symptoms)	Asymptotic (Lesion Found at MRI or Angiography without Consequent Symptoms)
**Dissections**	5	1	0	5
**Stenosis**	9	2	0	9
**Occusions**	3	2	2	1
**Hemorragic lesions**	5	5	1	4
**Lacunar Strokes**	38	38	20	18
**Strokes**	6	6	4	2
**Total of events on 3586 procedures**	66 (1.84%)	54 (1.5%)	27 (0.75%)	39 (1.08%)
**Focal Seizures (# of Events)**	**Generalized Seizures (# of Events)**	**Lymphomas**	**Metastasis**	**Glial Tumors**	**MTX**	**Carboplatin**
65	9	23	4	12	62	12
74 seizure events (2%)	39 patients (5.4%)
**Hematologic Toxicites (per NCIC Toxicity Criteria)**	**Grade 1**	**Grade 2**	**Grade 3**	**Grade 4**	**Total**
**Neutropenia**	70 (9.7%)	67 (9.3%)	22 (3.1%)	21 (2.9%)	180 (24.9%)
**Thrombocytopenia**	43 (5.9%)	37 (5.1%)	35 (4.9%)	21 (2.9%)	136 (18.8%)
**Anemia**	115 (15.9%)	78 (10.8%)	25 (3.5%)	10 (1.4%)	228 (31.6%)
**Total**	228 (31.6%)	182 (25.2%)	82 (11.4%)	52 (7.2%)

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
