# Peer review of "Drug Delivery Technology to the CNS in the Treatment of Brain Tumors: The Sherbrooke Experience"

_pharmaceutics, 2019, doi:10.3390/pharmaceutics11050248_

Round 1
Reviewer 1 Report
In this manuscript, the author provides an update on the “Sherbrooke experience”, i.e., the experience with certain CNS drug delivery strategies performed at Sherbrooke University in Canada. In particular, over the years this clinic has been using the intraarterial (IA) route as an avenue to deliver chemotherapeutics for the treatment of brain tumors, sometimes complemented by an osmotic breach of the blood brain barrier (BBB) with mannitol. Overall, the author presents clinical results with better responses than expected, along with low toxicity, in a large number of patients.
Altogether, this is an interesting manuscript, although two main issues minimize its value.
The first issue is an apparent lack of care in manuscript preparation. (i) There are numerous typos and grammatical issues, as well as inconsistent use of capitalization (GBM vs. gbm, Liquid chromatography vs. liquid chromatography, CNS Lymphoma vs. CNS lymphoma, etc.). Within Figure 3: “3rnd line” should be “3rd line”. Since there is only 1 author, “we present” should be “I present”. There are occasional changes in font size in the main text. (ii) The graphics and some of the text in several of the figures is too small and/or fuzzy. Parts of Figure 3 are indecipherable. Figure 5 is unclear to a non-expert; perhaps the addition of arrowheads would help in the left image; right image: please explain white vs. black areas. In Figure 6, font size in the top part of the figure is too small; it should be increased to match the font size of the bottom part of this figure. In Figure 8, the x-axis needs to be defined (presumably “months”). Thorough proofreading is advised.
In the “Intro” (please change to “Introduction”), the author states that “the last approved treatment in the management of primary brain tumour was the addition of temozolomide…” This is not quite correct, because tumor treating fields (TTF) also are approved for newly diagnosed GBM (in addition to recurrent GBM). The author further states that “Any further attempts to improve on these results [temozolomide] has been met with deception.” What is meant with “deception”? Please explain. Legend to Figure 9 mentions “Stupp regimen”; please add a reference (ref #1), as not everyone is familiar with this treatment.
While the above issue can be fixed easily, the second issue is more challenging. The manuscript appears to be an update (and extension) of a paper published in 2005 by the same author (and also subtitled “The Sherbrooke Experience.” In that previous paper, the authors also showed very encouraging results derived from intraarterial delivery during a Phase II study, and concluded that, based on their studies, a randomized Phase III study was opened. It is not clear what happened to this trial since then; it might be interesting to include an update. In the current study, transfemoral catherization is used to deliver chemotherapy, and this is “sometimes” complemented by an osmotic breach of the BBB with mannitol. It is not specified what “sometimes” means. Results are not separated into outcomes derived with or without mannitol. Does inclusion of IA mannitol result in a significant difference?
The manuscript harbors a combination of data from preclinical models (rats) and patients. As a review article, this would make sense; but this manuscript is not broad enough, nor has it sufficient explanations, to serve as a review. As a clinical report (about the Sherbrooke experience), it lacks patient selection criteria, patient demographics, and details regarding treatment results. Parts of the writing are very specialized and can be understood by neurosurgeons, radiologists, and the like (if presented in a typical neurosurgery journal). However, for a general audience (this journal is called “Pharmaceutics”), it would benefit from further descriptions. For example, Figure 5 is supposed to show “Clinical procedure: arterial catheterism”, but leaves the reader without an explanation what to look for in these images; nor does it actually describe the “procedure”. Similarly, mention of “Selective catherization via percutaneous transfemoral puncture” should be accompanied by a graphic or cartoon. It is not readily apparent to the non-specialized reader how specific areas of the brain can be reached by puncturing the groin area.
The data show good safety of this procedure; nonetheless, as the author mentions, its application remains restricted to specialized centers, which would preclude widespread use in general clinical practice—and thus qualifies the author’s assertion that “this approach is safe and sound”. As well, the author refers to the “extreme heterogeneity of GBM” and that the approach should be tailored “to each tumour for individual patient[s]”—which also would minimize general applicability in general practice. These aspects should be discussed more critically.
Achieving increased drug entrance into the brain does not necessarily result in increased survival (as the author has recently published with example of temozolomide in a rat model; Drapeau et al., 2017). Therefore, the preclinical data presented in the first half of this manuscript would benefit from a critical appraisal of this aspect. With regard to the clinical results that are shown in the second half: how do the data presented in Figure 8 (survival plot) compare to other clinically established treatments? As a reference, these curves could be compared to historical controls or, better yet, to the latest approved treatments for recurrent GBM, such as bevacizumab and tumor treating fields (neither of which is mentioned in this manuscript).
Author Response
All the comments of reviewer #1 were addressed. As to the fact that the paper is not a review, I entirely agree. Given the special edition of the journal discussing 'delivery technology in Canada', the paper simply aim to describe the work our laboratory has done in this field. It is mostly supported by our publications because of that. I hope that I understood correctly the scope of this special edition, and understand that this paper is neither a review, n ore an original experimental paper, because of that.
The other points addressed are as follows:
-All the figures have been edited and modified for clarity, based on the reviewer's comment.
-The 'intro' has been changed to introduction, and a reference and discussion of TTF were added.
-'deception' was changed for disappointing, and reference was added for the stupp regimen.
-A discussion was added in the clinical section related to the fact that the randomized studies has not started yet.
-A brief section was added to address the confusion of CIAC vs BBBD, and when each of these approaches are used. The reviewer was quite correct in pointing that this issue was confusing, and not properly addressed in the manuscript.
- The writing has been modified to further detail the clinical procedures for the non clinical readership. Several details and descriptions were added.
-'As well, the author refers to the “extreme heterogeneity of GBM” and that the approach should be tailored “to each tumour for individual patient[s]”—which also would minimize general applicability in general practice. These aspects should be discussed more critically'. This is an excellent point.
I have added a discussion on this issue
-'Achieving increased drug entrance into the brain does not necessarily result in increased survival (as the author has recently published with example of temozolomide in a rat model; Drapeau et al., 2017). Therefore, the preclinical data presented in the first half of this manuscript would benefit from a critical appraisal of this aspect.' Once again, a very good point made by the reviewer.
I have also added a brief discussion on this.
Reviewer 2 Report
The paper summarizes a relevant clinical experience. For sake of completeness some attention has been paid to pre-clinical studies. This paragraph needs some more work since figures are not completely described ( e.g. fig 1 part 2 is not commented) and also fig 3 and fig 4 and the related info are not explicative. ICP-MS has been performed in serum or in tumor tissue previously digested? At least some references should be given.
Also the way in which the pharmacokinetic data were obtained seems quite obscure
A careful reading is necessary because:
- somewhere Intraarterial infusion of chemotherapy is IAC somewhere else it is CIA,
-there are many grammatical errors :
line 27 remove 'on'
36: entity does not
70: remove of
74: lab has shown
89: help evading
129: concentration
150: continues
151: therapeutics
162: former
164: parameters such as
211: newly described cerebral lesions
216: peri-procedural
235: implies (better than comes with)
Author Response
A careful revision for synthax and abbreviation was performed.
All the figures were revised for clarity in accordance with this reviewer's comments.
Details were added regarding ICP-MS data.
Reviewer 3 Report
In this review article titled: " Drug delivery technology to the CNS in the treatment of brain tumours: the Sherbrooke experience". Dr. Fortin comprehensively addresses local CNS drug delivery strategies for malignant Brain Tumors (malignant gliomas and primary CNS lymphomas) which have no amenable cures (failure attributed to poor /none BBB penetration of therapeutics). Using the intraarterial (IA) route as an avenue to deliver chemotherapeutics in the treatment of brain tumors, and sharing the institutional clinical experiences on human patients the author has clearly demonstrated scientific rationale on the importance of this route of delivery, which represents a potential revolution in brain cancer treatment. Overall this is a timely and informative article as there are ongoing clinical trials using IA route of drug delivery for various brain malignancies.
Author Response
I have reviewed the paper for english. It was indeed submitted in a hurry, and benefited from a revision of english!
Round 2
Reviewer 1 Report
In this extensively revised manuscript, the author has significantly expanded the description of the relevant key clinical procedures, and he has provided several other details that nicely improve the significance and value of this manuscript. Overall, the content of this manuscript is remarkable and of interest.
Several of the previously mentioned issues with presentation and style were not adequately addressed, and therefore there remains room for improvement.
(i) There still are numerous typos and grammatical issues, along with inconsistent use of capitalization. (a) Typos: “pharmacokitenics”, “astroctyomas”, “intraarerial”, “angiography suit”, etc. (b) Subject/person/verb agreement: “in this papers”, “the IDH markers is”, “to avoid these heterogeneity”, “each therapeutics”, etc. (c) Inconsistent capitalization: “the Blood-brain barrier” should be “the blood-brain barrier”, “158 Metastasis” should be “158 metastasis”, “Mannitol” and “Melphalan” should be “mannitol” and “melphalan”; etc.. Line 96: The first letter in a title (or sentence) should be capitalized, and so should be trade names (“Lipoxal” instead of "lipoxal").
(ii) For professional presentation style, it would have been nicer to use similar font size for the numbers in Figure 6 top and bottom, with larger font size in the top part, matching the bottom part and making it easier to read. In Figure 8, “months” should be added to the x-axis in both panels.
(iii) As was mentioned, readers outside the immediate field of neuro-oncology might not be familiar with the term “Stupp regimen”. For that reason, a reference to this term (ref. #1) should be added to the legend of Figure 9. Better yet, please mention and explain this term in the Introduction, where the addition of temozolomide to radiotherapy is introduced.
(iv) In the context of temozolomide, the author has now added that “there is no clinical use of an IV formulation” of temozolomide (Line 268). This statement is not entirely correct and should be modified. In fact, oral temozolomide cannot be administered to all patients, in particular those with difficulties swallowing capsules, those with GI obstruction, and some others; here, IV temozolomide represents a viable alternative.
Author Response
All the typos and grammatical issues have been addressed.
The figures 6 and 8 have been corrected according to the reviewer' comments.
The Stupp regimen has been described in the introduction.
The statement on iv Temozolomide has been addressed.